# Plasma Concentrations of Matrilysins (MMP-7, MMP-26) and Stromelysins (MMP-3, MMP-10) as Diagnostic Biomarkers in High-Grade Serous Ovarian Cancer Patients

**DOI:** 10.3390/ijms26125661

**Published:** 2025-06-13

**Authors:** Gacuta Ewa, Paweł Ławicki, Hanna Grabowska, Michał Ławicki, Monika Kulesza, Aleksandra Kicman, Paweł Malinowski, Sławomir Ławicki

**Affiliations:** 1Department of Perinatology, University Clinical Hospital of Bialystok, 15-276 Bialystok, Poland; sunnyeve@wp.pl; 2Department of Population Medicine and Lifestyle Diseases Prevention, The Faculty of Medicine, Medical University of Białystok, 15-269 Bialystok, Poland; pawellawicki04@gmail.com (P.Ł.); grabowska.hanna@icloud.com (H.G.); mlawicki@icloud.com (M.Ł.); monika.kulesza@sd.umb.edu.pl (M.K.); 3Department of Psychiatry, The Faculty of Medicine, Medical University of Białystok, 15-272 Bialystok, Poland; olakicman@gmail.com; 4 Department of Oncological Surgery, Bialystok Oncology Center, 15-027 Bialystok, Poland; pawelmalinowski1981@gmail.com

**Keywords:** ovarian cancer, MMP-7, MMP-26, MMP-, MMP-10, HE4, CA125, matrilysins, stromelysins

## Abstract

Ovarian cancer (OC) has an extremely unfavourable prognosis. This is due to its asymptomatic course and lack of screening tests. Therefore, new methods are needed to diagnose OC. The aim of this study was to evaluate the concentrations and diagnostic utility of selected matrilysins and stromelysins in the diagnosis of OC in comparison with the classical markers CA125 and HE4. The study group included 100 patients with serous OC, 70 with serous cysts (BL), and 50 healthy women (HW). Selected MMPs were determined by ELISA, routine markers by CMIA. Ovarian cancer patients have elevated concentrations of MMP-7, MMP-26, MMP-10 as well as CA125 and HE4 in the total group and subgroups (stage I + II, and III + IV). The highest values of diagnostic parameters—SP, SE, NPV, PPV, and ACC, as compared to CA125 and HE4, were observed for MMP-7. Performing ROC analyses showed that the highest AUC values were observed for MMP-7, CA125, and HE4, in the whole group of patients and divided into stages I and II according to FIGO. Performing ROC analyses for groups III and IV according to FIGO was associated with an increase in AUC for the MMPs studied. Of the MMPs tested, MMP-7, MMP-26, and MMP-10 have the highest potential in diagnostics of serous ovarian cancer patients.

## 1. Introduction

Ovarian cancer is the eighth most common malignancy in postmenopausal women, accounting for less than one-third of all gynecological cancers. It is estimated that in 2020, it was responsible for 4.7% of all cancer-related deaths [1,2,3]. Ovarian cancer often remains asymptomatic for a long period of time, which is why it is frequently referred to as the “silent killer”. An additional factor complicating the early identification of patients with ovarian cancer is the absence of a specific, reliable screening procedure to detect this malignancy. The fact that ovarian cancer is typically diagnosed at an advanced stage contributes to a considerable deterioration in prognosis [3,4,5,6]. The type of High-Grade Serous Ovarian Carcinoma is the most commonly diagnosed type of and also has the worst prognosis. This type of cancer is responsible for 70–80% of ovarian cancer-related deaths [1].

The most commonly used and best understood diagnostic methods for ovarian cancer are transvaginal ultrasonography, imaging studies, and the measurement of biomarkers CA125 and HE4. However, none of these tests are sufficient as screening tools for ovarian cancer diagnosis. Data indicate that the CA125 and HE4 levels are elevated in the blood of ovarian cancer patients [7]. However, this method does not possess significant diagnostic value in the context of disease application. Furthermore, there may be a risk of false positives or limitations in the diagnostic utility of the CA125 and HE4 [2,3].

Early detection of localized ovarian cancer (stage I) is associated with a 90% cure rate. However, stages III and IV of ovarian cancer, where the disease has spread beyond the pelvis, are characterized by a survival rate of 20% or lower. Therefore, there is a compelling rationale for the exploration of new diagnostic methods, including cancer biomarkers, to improve the early detection of ovarian cancer [2,3,4,5]. Such methods include the determination of various compounds that have preliminary potential as biomarkers. These include, among others, extracellular matrix metalloproteinases.

These enzymes play a crucial role in tissue remodelling during various physiological processes, including angiogenesis, embryogenesis, morphogenesis, and wound healing. However, they are also implicated in pathological conditions such as myocardial infarction, fibrotic disorders, and osteoarthritis [2]. Matrix metalloproteinases (MMPs) have been extensively studied over the years, particularly for their potential role in carcinogenesis. Tissue overexpression and elevated blood levels of MMPs have been confirmed in a variety of malignancies, including prostate, lung, renal, colorectal, as well as common female cancers such as breast, ovarian, cervical, and endometrial cancer [2,4,5,8,9,10,11,12].

Accordingly, the purpose of this study was to determine the concentrations and diagnostic utility of selected matrilysins and stromelysins as novel markers of serous ovarian cancer in a comparison between women with benign lesions and healthy volunteers. The present study is a continuation of scientific research on the usefulness of MMPs as novel markers in the diagnosis of OC [4,5].

## 2. Results

### 2.1. Plasma Levels for Tested Parameters in Serous High-Grade Serous Ovarian Carcinoma Patients and in Control Groups

Plasma levels of tested parameters in High-Grade Serous Ovarian Carcinoma (OC) patients and in control groups were showed in Table 1 and Table 2 and in Figure 1, Figure 2, Figure 3, Figure 4, Figure 5 and Figure 6. Statistical analysis showed that ovarian cancer patients in the total group had significantly higher levels of most of all of the tested MMPs (median MMP-7: 4.48 ng/mL; MMP-26: 9085.0 ng/mL; MMP-10: 95.45 ng/mL) as well as CA125 (313.55 U/mL) and HE4 (131.35 U/mL), compared to healthy women group (0.95 ng/mL; 7162.5 pg/mL; 49.54 ng/mL; 16.9 U/mL; 39.51 U/mL; *p* < 0.0001; *p* < 0.0001; *p* < 0.0001; *p* < 0.0001; *p* < 0.0001, respectively). Furthermore, also statistically higher concentrations of all tested MMPs were observed in cystis serous patients in comparison to the healthy women group: MMP-7 (2.96 ng/mL), MMP-26 (10,285 pg/mL), MMP-10 (209.438 pg/mL), CA125 (median 23.75 IU/mL), and HE4 (51.30 IU/mL), (*p* < 0.0001; *p* = 0.772; *p* < 0.0001; *p* < 0.0001; *p* < 0.0001, respectively). Only MMP-3 concentrations were comparable between study groups (OC median: 9190.8 ng/mL; BL: 9367.86; HW median: 8359.42 ng/mL). There were no statistically significant differences between MMP-3 concentrations in the OC and BL groups (*p* = 0.732). Also for MMP-26, the analysis showed no differences between the concentrations of this enzyme in the OC and BL groups (*p* = 0.772) but for the other MMPs tested, higher concentrations of these enzymes were obtained in the OC patient group compared to BL (MMP-7: 2.9 ng/mL), *p* < 0.0001; MMP-10 (49.54 ng/mL; *p* < 0.0001).

The serous OC group was also tested for parameters divided on the basis of the stage of ovarian cancer according to FIGO classification. Patients were divided into two groups: group I—FIGO stages I and II, and group II—FIGO stages III and IV.

MMP-7 concentrations were slightly higher in stage III–IV group (4.8 ng/mL) than in stage I–II group (4.5 ng/mL). In contrast, MMP-26 concentrations were comparable in both groups (I–II: 9240 pg/mL; III–IV: 9220 pg/mL). For MMP-3 (9328 pg/mL), MMP-10 (103.8 ng/mL), and CA125 (495.7 U/mL), higher concentrations are found in stages III–IV compared to stages I–II (resp. 9065 pg/mL; 87.06 ng/mL; 105.1 μ/mL). HE4 concentrations were higher in stages I–II (182.4 U/mL) compared to III–IV (140.4 U/mL). This is shown in Table 3. At the same time, higher concentrations of MMP-7, MMP-26, MMP-10, CA125, and HE4 are found in both stages (I–II and III–IV) compared to the group of healthy women. In addition, higher concentrations of the tested markers (except for MMP-3) were found in both subgroups of serous ovarian cancer compared to benign lesions (see Table 3).

### 2.2. Correlation by Spearman’s Method

Spearman’s method was used to analyze potential correlations. The analysis revealed both positive and negative correlations; however, only in the ovarian cancer group were they statistically significant. Statistically significant positive correlations in the ovarian cancer group were observed between MMP-26 and MMP-10 (r = 0.2168; *p* = 0.0101). MMP-26 and CA125 (r = 0.1725; *p* = 0.0415), MMP-10 and MMP-3 (r = 0.1963; *p* = 0.0201), and MMP-10 and HE4 (r = 0.2131; *p* = 0.0115). Statistically significant negative correlations were observed for MMP-7 and MMP-26 (r = −0.2519; *p* = 0.0119); MMP-7 and MMP-10 (r = −0.2030 *p* = 0.0161); MMP-3 and MMP-26 (r = −0.2086; *p* = 0.0134). r = −0.2164 *p* = 0.0102); and MMP-26 and HE4 (r = −0.2164; *p* = 0.0102). The results are shown in Table 3. 

### 2.3. Diagnostic Criteria of Tested MMPs: CA 125 and HE-4

Table 4 and Table 5 contains the diagnostic criteria—diagnostic sensitivity (SE) and diagnostic specificity (SP), positive and negative predictive value (PPV and NPV), and accuracy (ACC) in serous ovarian cancer patients’ total group.

The highest SE for the serous ovarian cancer patient total group was achieved by MMP-7 (94%), which was the only MMP that slightly overtook CA125 (93%). The SE value of HE-4 (75%) was surpassed by MMP-7 (94%), MMP-10 (81%), MMP-26 (82%), and also CA125 (93%). SP values ranged between 60% and 86% and did not surpass CA125 (98%) or HE-4 (94%). The PPV had high values (76–93%) for all the parameters, but the highest level was reached by CA125 (98.94%). The second highest level was observed for HE-4 (96.15%). The highest value of NPV was observed for MMP-7 (87.76%), which was only slightly higher than CA125 (87.5%). The accuracy for tested parameters was between 62.67 and 94.67%. The highest value was observed for CA125, and the second highest value for MMP-7 (91.33%).

MMP-7 and MMP-10 have the highest sensitivity values in stage I–II (both 91.67%). The highest specificity values were also achieved for these MMPs (MMP-7: 86%; MMP-10: 62%). For PPV, the best result of 75.86% was obtained for MMP-7, while NPV was also the highest for MMP-7 at 95.56%. ACC for MMP-7 was also the highest. reaching the value of 91.33%. For CA125 and HE4, high values of diagnostic parameters were achieved—CA125 (SE: 93%; SP: 98%; PPV: 98.94%; NPV: 87.5%; and ACC: 94.67%) and HE4 (SE: 75% SP: 94% PPV: 96.15% NPV: 65.28%; and ACC: 81.33%).

For stage III–IV, the best values of diagnostic parameters were obtained for MMP-7 with a sensitivity value of 96%, specificity of 86%, positive predictive value of 71.01%. negative predictive value of 54.55%, and diagnostic accuracy of 63.71%. Also, in the stage III–IV subgroup, high values of diagnostic parameters for CA125 and HE4 were observed—CA125 (SE: 97.33% SP: 98.00% PPV: 98.65% NPV: 96.08% and ACC: 97.60%) and HE4 (SE: 73.33% SP: 94.00% PPV: 94.83% NPV: 70.15% and ACC: 81.60%).

### 2.4. Evaluation of the Diagnostic Power of Tests by ROC Function

The diagnostic power of the tests was evaluated by analyzing the area under the ROC curve. Performing such an analysis makes it possible to distinguish normal results from abnormal ones. A diagnostically ideal test makes it possible to completely distinguish a healthy person from a sick person, with a sensitivity value of 100% and specificity of 100%, where the line in the ROC function will completely coincide with the Y axis, and the AUC will reach a value of 1. An unusable test does not make it possible to distinguish sick people from healthy people. Such a test will be in the form of a straight line inclined at 45 degrees to the X axis. The AUC value will be close to 0.5, which is the limit of the diagnostic usefulness of the test. The results of the ROC curve analysis are shown in Table 6.

Among the MMPs tested. the highest AUC value in the total tested group was obtained for MMP-7 (0.9566). This value exceeded the AUC value obtained for HE4 (0.9136). The remaining MMPs acquired high AUC values but did not exceed the AUC values for routine markers. The AUC (0.5821) value obtained for MMP-3 was the lowest. but it still exceeded 0.5. All tests exceeded AUC = 0.5. which means they are diagnostically useful. Figure 7 shows the ROC curves for the parameters studied.

ROC analysis was also performed for parameters divided based on the stage of ovarian cancer, according to the FIGO classification: stages I and II and III and IV groups. Table 7 shows ROC curve analyses, obtained results are for ovarian cancer patients divided into stages I–II and III–IV.

Performing a breakdown by FIGO stage I–II, again, the highest AUC values were obtained for MMP-7 (0.9548) as well as CA125 (0.9840) and HE4 (0.9179). The lowest AUC value was obtained for MMP-3 (0.5865). Figure 8 and Figure 9 shows the ROC curves obtained for the studied parameters with the division according to the FIGO classification by stage I and II.

The highest AUC was observed for CA125 (0.9955 and MMP-7 (0.9574). The AUC values for MMP-10 (0.8065) and MMP-26 (0.7548) exceeded the AUC values for the routine marker HE4 (0.6843). Additional materials for the work have been placed in Appendix A. 

## 3. Discussion

Currently, diagnosis of OC is based on transvaginal ultrasonography (TVS) and determination of tumour markers—CA125 and/or HE4. These markers, despite their good sensitivity and specificity, are not suitable for diagnosing this condition. It is unfortunate that the number of ovarian cancer cases is steadily increasing. It is estimated that in 2050 the number of cases will reach 75,570. In comparison, 46,232 cases of the condition were detected in 2022 [5]. Additionally, the majority of patients are diagnosed at high stages of the disease, i.e., stage III or IV. This further translates into an unfavourable prognosis for patients with the condition. Importantly, OC is most often asymptomatic. Patients rarely report non-specific symptoms such as lower abdominal pain, frequent urinary urgency, or general fatigue [2,3,4,13]. Given the ever-increasing incidence, the lack of screening tests, and the asymptomatic course, new methods should be sought to enable faster diagnosis of OC. Indeed, it should be emphasized that earlier detection of OC translates into a better prognosis for female patients [2,3,4,5,6]. At present, there are many potential compounds proposed as potential markers, but matrix metalloproteinases have exceptional potential.

Therefore, the aim of this study was to evaluate the concentrations and diagnostic usefulness of selected enzymes from the matrilysin group (MMP-7, MMP-26) and stromelysins (MMP-3 and MMP-10) in the most common histological type of ovarian cancer—the serous type. Compared to patients with benign lesions and healthy women. This work is a continuation of our team’s previous studies on the usefulness of MMPs in the diagnosis of ovarian cancer [4,5,14].

MMP-7 belongs to the matrilysin group. It is a small enzyme that has been shown to play an important role in the course of carcinogenesis where it mediates the proliferation. Differentiation, and invasion of tumour cells. Additionally, it is involved in metastasis formation [15]. Expression of this enzyme is found in normal ovarian tissue [2,16,17] and in ovarian cancer tumour tissue [2,16,18]. Interestingly, this expression was found both directly in tumour tissue, in the stroma surrounding the tumour, and in metastatic foci [19]. Importantly, this expression was higher in serous ovarian cancer compared to benign lesions and healthy patients [16]. This is also consistent with our study, where we found higher levels of MMP-7 (4.48 ng/mL) compared to benign lesions (2.96 ng/mL; *p* < 0.0001) and healthy women (0.95 ng/mL; *p* < 0.0001). This is also consistent with the studies of Acar et al. [20], Gershtein et al. [21], Meinhold-Heerlein et al. [22], and Będkowska et al. [14] who found higher concentrations of MMP-7 in ovarian cancer patients compared to women with benign lesions and healthy women. Importantly, the concentrations in the work of Meinhold-Heerlein et al. [22] (OC: about 5 ng/mL; BL about 2 ng/mL and 1 ng/mL) were comparable to the medians obtained in our study. In the study by Acar et al. [20], concentrations in patients with OC MMP-7 were higher (10.24 +/− 1.35 ng/mL) than in healthy women (3.29 +/− 1.64 ng/mL) than in our study, but this was probably due to the greater diversity of the group; the study included patients with different histological types of OC. In addition, the team of Acar et al. [20] performed tests on serum. Moreover, our study showed a difference in MMP-7 levels between subgroups divided according to FIGO; patients in higher stages had slightly higher levels of this enzyme than patients in lower stages. This may indicate that tumours in more advanced stages produce more MMP-7.

MMP-26 also belongs to the matrilysin group and, like MMP-7, is an important enzyme in the progression of cancer. Studies have shown that MMP-26 is involved in growth, invasion, and angiogenesis within the tumour, thereby contributing to tumour growth and progression [23]. MMP-26 is expressed in the normal ovary, and MMP-26 expression is also found in OC cancerous tissue [2,24]. In our study, MMP-26 levels were higher than in healthy women (OC 9085 pg/mL; HS 7162.5 pg/mL) and at the same time lower than in women with benign lesions (10,285 pg/mL). This agrees with the study of Kicman et al. [4], who obtained similar results (OC: 9330 pg/mL; BL: 10,440 pg/mL and HS: 7162.5 pg/mL). In this team’s work, women with benign lesions also had the highest levels of MMP-26 compared to the other patient groups. It should be noted, however, that this research team qualified patients with different histological types of ovarian cancer. MMP-3 belongs to the group of stromelysins, and its important role in tumour cell differentiation and proliferation, as well as in angiogenesis has been demonstrated [25]. Interestingly, in the case of MMP-26, we did not find differences in concentrations between stages I–II and III–IV, which indicates that the production of MMP-26 by cancer cells does not depend on the stage of cancer and remains at a constant level.

MMP-3 expression is found in normal ovarian tissue [26] and in tumour tissue [2]. MMP-3 expression was found in all histological types of ovarian cancer. It should be noted that MMP-3 expression in ovarian cancer was associated with the degree of disease according to FIGO—patients in higher grades had higher expression of this enzyme [2]. According to our study, ovarian cancer patients had slightly higher levels (9190.80 pg/mL) compared to healthy women (8359.42 pg/mL). MMP-3 concentrations in ovarian cancer patients and women with benign lesions (9367.86 pg/mL). These results agree with the work of Kicman et al. [4], who found comparable MMP-3 concentrations in OC and BL patients (OC^−^ 9.32 ng/mL; BL 9.84 ng/mL) and higher MMP-3 concentrations in OC patients compared to healthy women (7.97 ng/mL). It should be noted that MMP-3 concentrations in healthy women in our study and in the study by Kicman et al. [4] are comparable. At the same time, our findings do not agree with those of Cymbaluk-Ploska et al. [27], who found significantly higher MMP-3 concentrations in OC patients compared to BL (OC: 14.657 ng/mL; BL 9.84 ng/mL). This difference may be due to the different method of MMP-3 determination and the greater variety of forms of benign ovarian lesions—only patients with serous lesions were eligible to participate in our study. As with MMP-7, MMP-3 concentrations were higher in stage III–IV than in lower stages. This is likely related to the size and dynamics of the ovarian cancer tumour.

The last enzyme assayed by our team is MMP-10, which belongs to the stromelysin group. This enzyme is involved in the breakdown of extracellular matrix, thereby contributing to metastasis formation. It also mediates anigogenesis [28]. Importantly, MMP-10 expression is not found in physiological ovarian tissue; it is only confirmed in tumour tissue [2,18]. The appearance of mRNA for MMP-10 may be associated with the neoplastic process. We are now the first research team to determine plasma levels of MMP-10 in ovarian cancer patients. In our study, we have shown that women with benign lesions have the highest concentrations of all the groups studied (OC 95.45 ng/mL; BL^−^ 209.44 ng/mL; 49.54 ng/mL). These results suggest the preliminary potential of MMP-10 as a negative marker in the differential diagnosis of benign from malignant lesions. It can be especially useful as an adjunct test to the ROMA algorithm. Also, MMP-10 concentrations were higher in group III–IV compared to group I–II, which suggests that, as with MMP-7 and MMP-3, this enzyme is produced at higher levels in advanced stages of cancer.

Our study showed higher levels of both CA125 and HE4 in ovarian cancer patients (CA125: 313.55 U/mL; HE4 131.35 U/mL) compared to BL women (CA125 23.75 U/mL; HE4: 51.30 U/mL) and healthy women (CA125: 16.9 U/mL; HE4: 39.51 U/mL). This agrees with the results of Kicman et al. [4] and Kicman et al. [5]. This indicates that the methodology of our study was correct. In our study, after dividing patients into stages I–II and III–IV, we found higher CA125 levels in more advanced stages. On the other hand, HE4 levels were higher in patients in lower stages. This indicates that both markers have diagnostic potential in different stages of serous ovarian cancer—CA125 in higher, while HE4 in earlier stages.

A series of analyses—sensitivity (SE), specificity (SP), negative (NPV) and positive (PPV) predictive value, diagnostic accuracy (ACC), and test power—were performed to determine the diagnostic usefulness of the tested parameters. The highest values of the tested parameters were obtained for routine markers. However, it should be noted that despite good values of diagnostic parameters and definitely higher concentrations in OC patients, these markers have numerous limitations, which include interference with associated pathological and physiological conditions in the female body [29,30]. Of the MMPs tested, MMP-7 (SE: 94%; SP: 86%; PPV: 93.07%; NPV: 87.76%; and ACC: 91.33%) and MMP-26 (SE: 82%; SP: 60%; PPV: 80.39%; and ACC: 74.67%) have the best diagnostic performance in total OC group and in stages I–II and III–IV. However, further studies are needed on the potential usefulness of MMPs in the diagnosis of ovarian cancer.

## 4. Materials and Methods

### 4.1. Patients

In this study, we included a group of 100 patients with High-Grade Serous Ovarian Carcinoma who underwent diagnosis and subsequent surgical treatment at the University Clinical Hospital in Bialystok. The ovarian cancer histopathology was established in all cases by tissue biopsy of the tumour or after surgery treatment from the tumour cancer tissues. None of the patients had received chemo- or radiotherapy before blood sample collection. Ovarian cancer patients participating in the study were divided into groups according to the stage of ovarian cancers (I–IV stage) following the FIGO classification. The control groups consisted of 73 subjects with benign ovarian tumour patients (cystis serous) and 50 healthy women. The benign ovarian tumour histopathology was established in all cases by tissue biopsy of the ovarian tumour or after surgery. All patients within the tested and control groups were also checked to see if they were pre- or postmenopausal. Patients with comorbidities such as cardiovascular disease, infections, and other types of cancer were excluded from the study. Patients who had previously received chemotherapy or radiation therapy and minors were also ineligible for the study. Due to interference with HE4 concentrations, patients with renal failure were also not qualified for the study. Characteristics of the studied groups are presented in Table 8.

We qualified patients with ovarian cancer or a benign ovarian lesion (cystis serous) based on gynecological examinations followed by confirmatory examinations by the oncologist based on imaging studies (USG/magnetic resonance imaging) and laboratory tests. The healthy women included in the control group were volunteers who were qualified to participate in the study by a family doctor and then a gynecologist of the University Clinical Hospital in Bialystok, and participants of the Bialystok PLUS cohort study. A detailed imaging diagnosis (abdominal or intravaginal ultrasound/magnetic resonance) and evaluation of laboratory results were performed. On the basis of these, the gynecologist subsequently determined the subjects’ suitability for being included in the study. The study was conducted in accordance with the Declaration of Helsinki and approved by the Ethical Committee of the Medical University of Bialystok (APK.002.420.2021 [approval dates 21 October 2021 and 19 December 2024).

### 4.2. Biochemical Analyses

The study material was plasma obtained from venous blood collected for the anticoagulant—lithium heparin. Venous blood was collected from the participants and centrifuged at 1810× *g* for 10 min. The centrifuged plasma was then pooled and stored at −85 °C until the day of the assay.

We measured plasma matrilysin (MMP-7, MMP-26) and stromelysin (MMP-3, MMP-10) concentrations with the use of the immuno-enzymatic ELISA method (Quantikine ELISA Human. R&D Systems Inc., Minneapolis, MN, USA). The assays were performed according to the manufacturer’s instructions provided with the kits, using double sample determinations for the standard curve and the tested samples. For measurement of CA 125 and HE-4 levels we used a chemiluminescent microparticle immunoassay (CMIA) (Abbott, Chicago, IL, USA) according to the manufacturer’s protocols.

### 4.3. Statistical Analysis

The parameter analysis was performed using IBM SPSS Statistics for Windows, version 29.0.2.0 (IBM Corp, Armonk, NY, USA). Calculations related to the ROC curve were performed with the Medcalc programme.

After the evaluation of the normality of the distribution for the tested parameters with the Shapiro–Wilk test with Liliefors correction, which revealed significant deviations from the normal distribution, we performed a statistical analysis using nonparametric tests. To assess statistical differences between independent groups, we used the Kruskal–Wallis test. and the Mann–Whitney U test. whereas when comparing multiple groups. we used the Mann–Whitney U test with Holm–Bonferroni correction for multiple inquiries or the Dwass–Steele–Critchlow–Fligner post hoc test.

Medcalc’s Free Statistical Calculators were used to evaluate the diagnostic features of the single parameters: sensitivity (SE), specificity (SP), positive predictive value (PPV), negative predictive value (NPV), and accuracy (ACC). Our analysis was performed based on the area under the ROC curve (AUC) and the optimal cut-off points determined with the maximizing value of the Youden index for the cancer–control differentiation test, which were 2.197 ng/mL (MMP-7), 7550 pg/mL (MMP-26), 8803.23 pg/mL (MMP-3), 59.58 pg/mL (MMP-10), and 30.185 IU/mL for CA 125 and HE4 (67.11 IU/mL).

## 5. Conclusions

The tested matrilysins MMP-7 and MMP-26 and stromelysin MMP-10 have the highest initial diagnostic potential in detecting High-Grade Serous Ovarian Carcinoma.

## Figures and Tables

**Figure 1 ijms-26-05661-f001:**
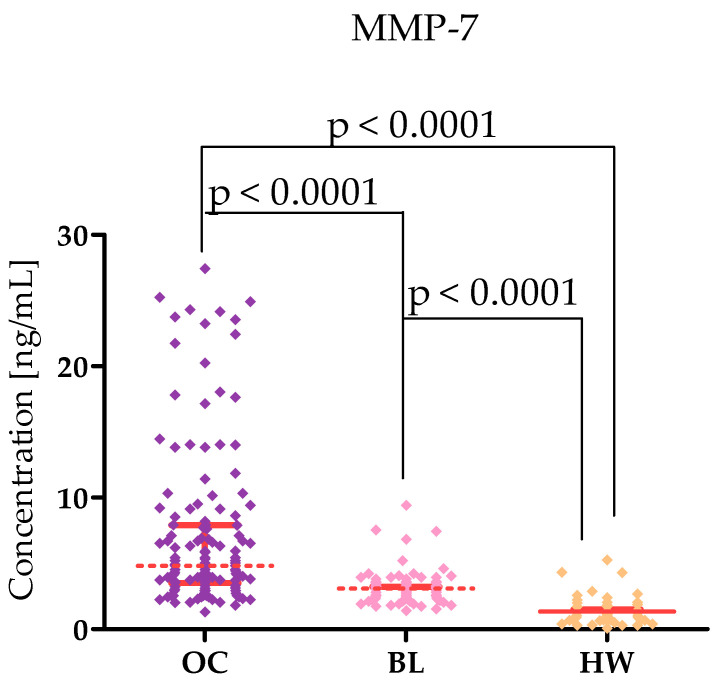
MMP-7 levels in patients with High-Grade Serous Ovarian Carcinoma (OC—total group), cysts (BL), and healthy women (HW) with median and interquartile range and *p* value.

**Figure 2 ijms-26-05661-f002:**
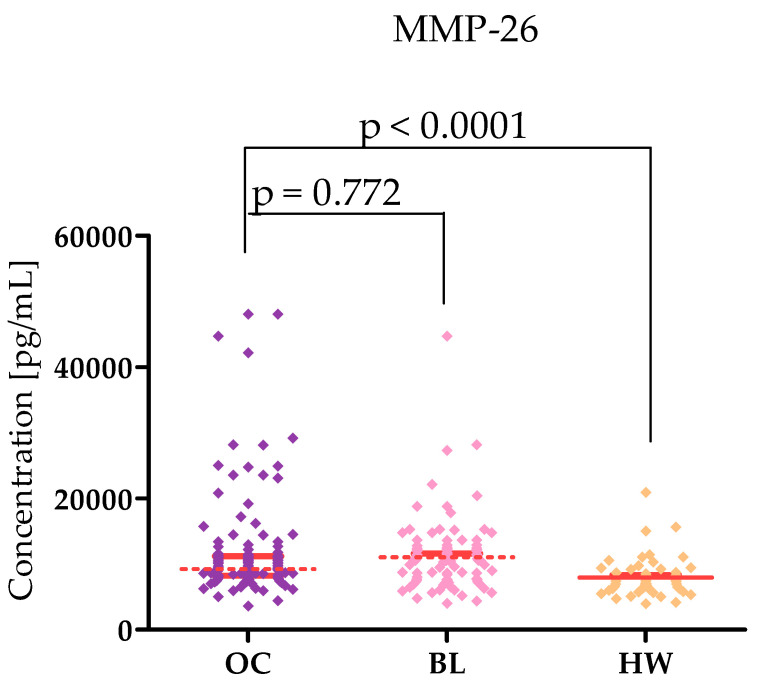
MMP-26 levels in patients with High-Grade Serous Ovarian Carcinoma (OC—total group), cysts (BL), and healthy women (HW) with median and interquartile range and *p* value.

**Figure 3 ijms-26-05661-f003:**
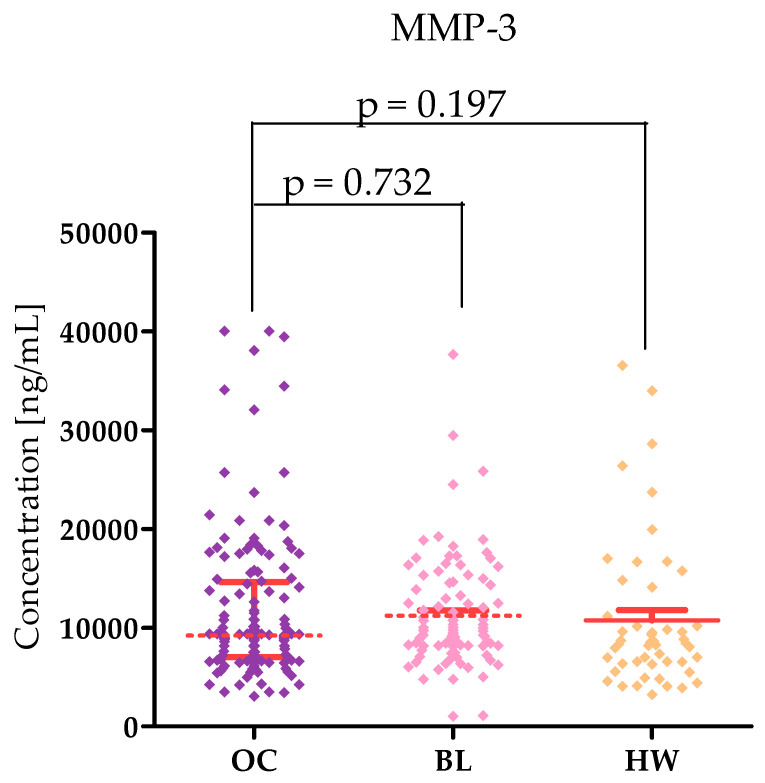
MMP-3 levels in patients with High-Grade Serous Ovarian Carcinoma (OC—total group), cysts (BL), and healthy women (HW) with median and interquartile range and *p* value.

**Figure 4 ijms-26-05661-f004:**
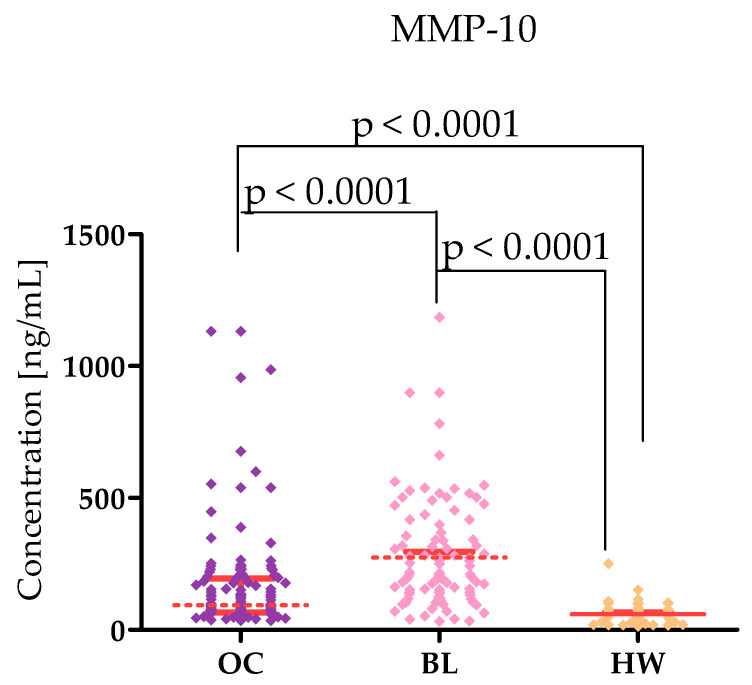
MMP-10 levels in patients with High-Grade Serous Ovarian Carcinoma (OC—total group), cysts (BL), and healthy women (HW) with median and interquartile range and *p* value.

**Figure 5 ijms-26-05661-f005:**
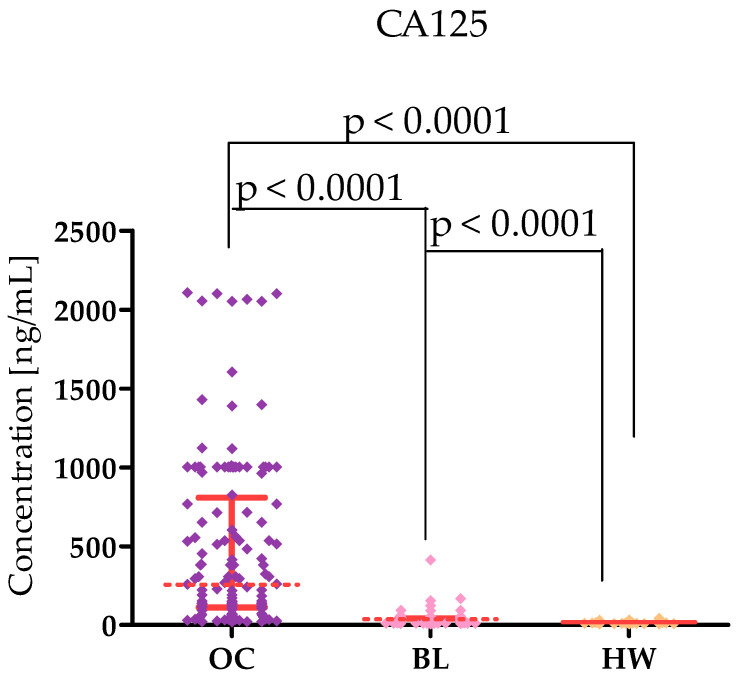
CA125 levels in patients with High-Grade Serous Ovarian Carcinoma (OC—total group), cysts (BL), and healthy women (HW) with median and interquartile range and *p* value.

**Figure 6 ijms-26-05661-f006:**
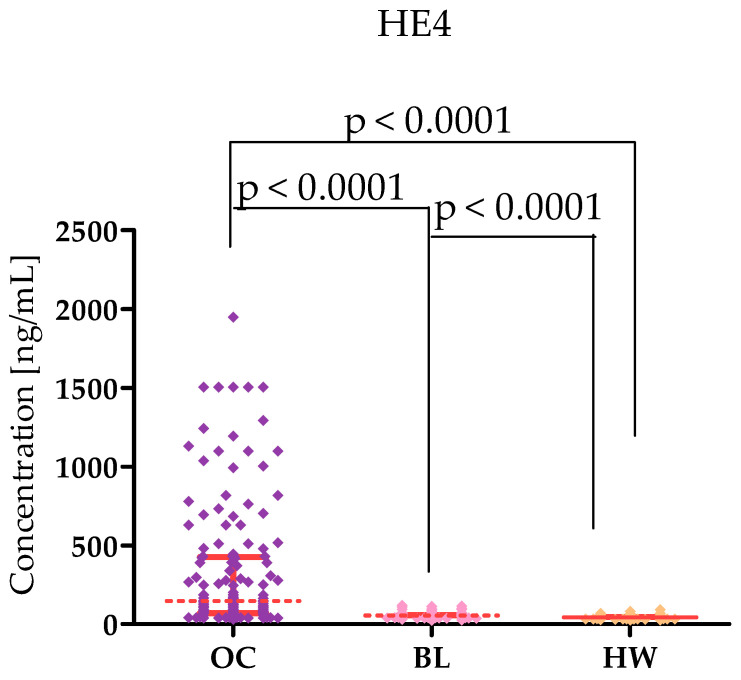
HE4 levels in patients with High-Grade Serous Ovarian Carcinoma (OC—total group), cysts (BL), and healthy women (HW) with median and interquartile range and *p* value.

**Figure 7 ijms-26-05661-f007:**
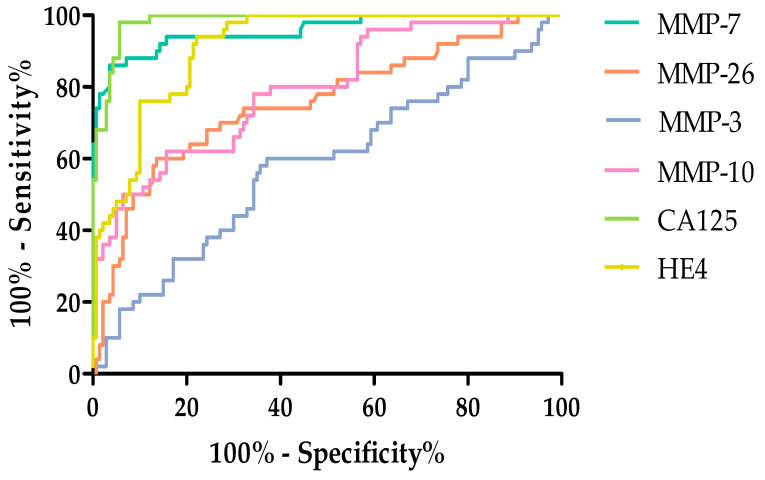
ROC curve for tested parameters in serous OC total group.

**Figure 8 ijms-26-05661-f008:**
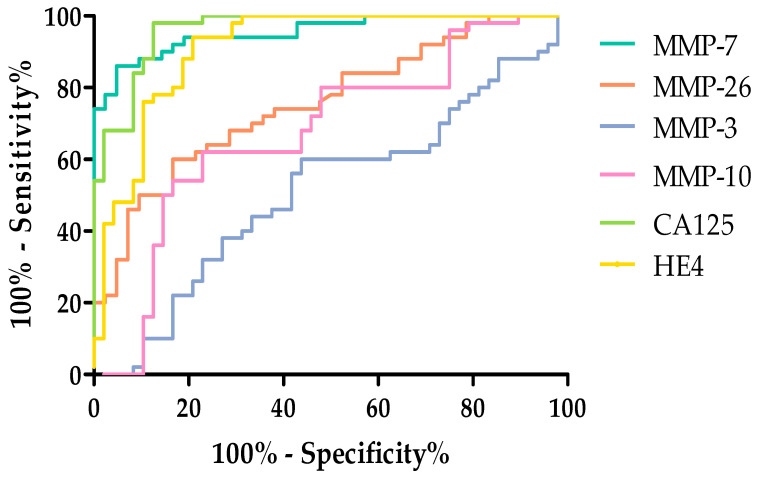
ROC curve for the studied parameters in stages I and II according to FIGO classification.

**Figure 9 ijms-26-05661-f009:**
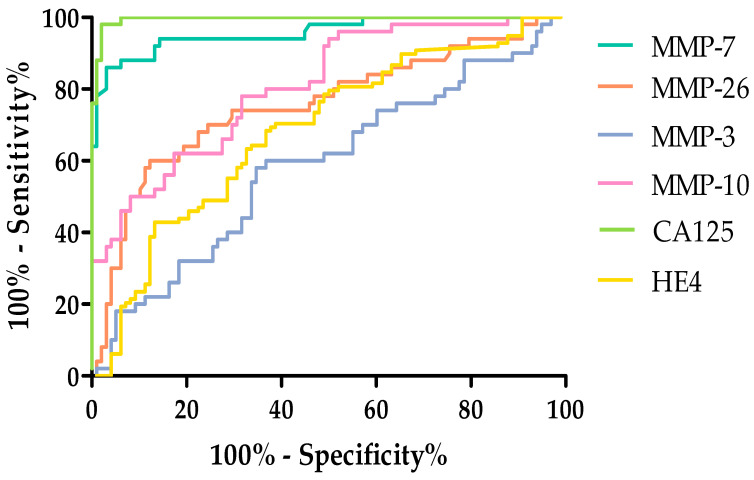
ROC curve for the studied parameters in stages III and IV according to FIGO classification.

**Table 1 ijms-26-05661-t001:** Plasma levels for tested MMPs, CA125, and HE4 in serous OC divided into stage I–II and III–IV groups.

Tested Parameters	Ovarian Cancer (OC) Stage I–II Median Range 95 Percentile 5 Percentile	Ovarian Cancer (OC) Stage III–IV Median Range 95 Percentile 5 Percentile	Statistical Significance
MMP-7 [ng/mL]	4.500 1.800–27.40 24.28 2.002	4.800 1.300–25.20 2.200 20.20	*p* < 0.0001
MMP-26 [pg/mL]	9240 5850–47,950 44,383 6092	9220 3578–47,950 23,455 6175	*p* < 0.0001
MMP-3 [pg/mL]	9065 4159–39,441 25,378 4224	9328 3033–40,000 34,304 4179	*p* = 0.197
MMP-10 [ng/mL]	87.06 38.90–1131 566.3 50.29	103.8 33.60–1131 551.4 41.40	*p* < 0.0001
CA125 [U/mL]	105.1 17.10–1000 411.6 19.46	495.7 23.00–4113 2063 38.82	*p* < 0.0001
HE4 [U/mL]	182.4 28.90–1944 1222 36.35	140.4 35.57–1500 1500 37.93	*p* < 0.0001

**Table 2 ijms-26-05661-t002:** Spearman’s correlation for the tested parameters.

Parameter	MMP-7	MMP-26	MMP-3	MMP-10	CA125	HE4
Serous Ovarian Cancer						
MMP-7	-	r = −0.2519 *p* = 0.0119	r = −0.1741 *p* = 0.0849	r = −0.08830 *p* = 0.3848	r = 0.02867 *p* = 0.9775	r = 0.07760 *p* = 0.4452
MMP-26	r = 0.02042 *p* = 0.8107	-	r = −0.2086 *p* = 0.0134	r = −0.2312 *p* = 0.0060	r = 0.1725 *p* = 0.0415	r = −0.2164 *p* = 0.0102
MMP-3	r = −0.06679 *p* = 0.4330	r = −0.02095 *p* = 0.8060	-	r = 0.1963 *p* = 0.0201	r = 0.07582 *p* = 0.3732	r = 0.04334 *p* = 0.6112
MMP-10	r = −0.2030 *p* = 0.0161	r = 0.2168 *p* = 0.0101	r = 0.1963 *p* = 0.0201	-	r = 0.07967 *p* = 0.3494	r = 0.2131 *p* = 0.0115
Benign lesion						
MMP-7	-	r = 0.1666 *p* = 0.0993	r = 0.03748 *p* = 0.7126	r = −0.1047 *p* = 0.3025	r = −0.04681 *p* = 0.6455	r = 0.02135 *p* = 0.8339
MMP-26	r = 0.1666 *p* = 0.0993	-	r = 0.126 *p* =0.952	r = 0.03624 *p* = 0.7218	r = 0.05994 *p* = 0.9530	r = −0.1387 *p* = 0.1709
MMP-3	r = 0.03748 *p* = 0.7126	r = 0.05994 *p* = 0.9530	-	r = 0.1118 *p* = 0.2704	r = 0.04293 *p* = 0.6731	r = −0.06714 *p* = 0.5090
MMP-10	r = −0.1047 *p* = 0.3025	r = 0.03624 *p* = 0.7218	r = 0.1118 *p* = 0.2704	-	r = −0.05854 *p* = 0.5649	r = −0.1463 *p* = 0.1486
Healthy women						
MMP-7	-	r = 0.1145 *p* = 0.4284	r = −0.2479 *p* = 0.0825	r = 0.2146 *p* = 0.1346	r = 0.2308 *p* = 0.1069	r = 0.1665 *p* = 0.365
MMP-26	r = 0.1145 *p* = 0.4284	-	r = 0.07582 *p* = 0.6007	r = −0.1023 *p* = 0.4797	r = −0.03011 *p* = 0.9836	r = 0.07276 *p* = 0.6193
MMP-3	r = −0.2479 *p* = 0.0825	r = 0.07582 *p* = 0.6007	-	r = 0.1427 *p* = 0.3228	r = 0.03492 *p* = 0.8098	r = 0.08869 *p* = 0.5402
MMP-10	r = 0.2146 *p* = 0.1346	r = −0.1023 *p* = 0.4797	r = 0.1427 *p* = 0.3228	-	r = −0.1389 *p* = 0.3362	r = −0.1787 *p* = 0.2144

Red indicates a statistically significant correlation.

**Table 3 ijms-26-05661-t003:** Diagnostic criteria for tested parameters individually in the group including all patients with serous ovarian cancer.

		SE	SP	PPV	NPV	ACC
MMP-3	value [%]	64%	60%	76.2%	45.45%	62.67%
	95 cl [%]	53.79–73.36%	45.18–73.59%	68.85–82.25%	37.10–54.08%	54.40–70.42%
MMP-7	value [%]	94%	86%	93.07%	87.76%	91.33%
	95 cl [%]	87.40–97.77%	73.26–94.18%	87.09–96.40%	76.60–94.01%	85.64–95.30%
MMP-10	value [%]	81%	62%	81%	62%	74.67%
	95 cl [%]	71.93–88.16%	47.17–75.35%	74.71–86.02%	50.76–72.09%	66.93–81.41%
MMP-26	value [%]	82%	60%	80.39%	62.5%	74.67%
	95 cl [%]	73.05–88.97%	45.18–73.59%	74.26–85.35%	50.88–72.84%	66.93–81.41%
CA125	value [%]	93%	98%	98.94%	87.5%	94.67%
	95 cl [%]	86.11–97.14%	89.35–99.95%	93.03–99.85%	77.39–93.47%	89.76–97.67%
HE-4	value [%]	75%	94%	96.15%	65.28%	81.33%
	95 cl [%]	65.34–83.12%	83.45–98.75%	89.24–98.69%	57.07–72.67%	74.16–87.22%

SE: Sensitivity; SP: Specificity; PPV: Positive predictive value; NPV: Negative predictive value; ACC: Diagnostic accuracy.

**Table 4 ijms-26-05661-t004:** Diagnostic criteria of tested parameters individually in patients with serous ovarian cancer in stages I–II.

Stage I–II						
		SE	SP	PPV	NPV	ACC
MMP-3	value [%]	58.33%	60.00%	41.18%	75.00%	59.46%
	95 cl [%]	36.64–77.89%	45.18–73.59%	30.24–53.06%	63.97–83.53%	47.41–70.73%
MMP-7	value [%]	91.67%	86.00%	75.86%	95.56%	87.84%
	95 cl [%]	73.00–98.97%	73.26–94.18%	61.01–86.33%	85.02–98.79%	78.16–94.29%
MMP-10	value [%]	91.67%	62.00%	53.66%	93.94%	71.62%
	95 cl [%]	73.00–98.97%	47.17–75.35%	44.34–62.73%	80.16–98.35%	59.95–81.50%
MMP-26	value [%]	75.00%	60.00%	47.37%	83.33%	64.86%
	95 cl [%]	53.29–90.23%	45.18–73.59%	37.38–57.57%	70.69–91.20%	52.89–75.61%
CA125	value [%]	83.33%	98.00%	95.24%	92.45%	93.24%
	95 cl [%]	62.62–95.26%	89.35–99.95%	74.02–99.29%	83.34–96.77%	84.93–97.77%
HE-4	value [%]	83.33%	94.00%	86.96%	92.16%	92.16%
	95 cl [%]	62.62–95.26%	83.45–98.75%	68.69–95.30%	82.73–96.65%	81.48–96.11%

**Table 5 ijms-26-05661-t005:** Diagnostic criteria of tested parameters individually in patients with serous ovarian cancer in stages III–IV.

Stage III–IV						
		SE	SP	PPV	NPV	ACC
MMP-3	value [%]	66.22%	60.00%	71.01%	54.55%	63.71%
	95 cl [%]	54.28–76.81%	45.18–73.59%	62.71–78.12%	44.80–63.96%	54.60–72.15%
MMP-7	value [%]	96.00%	86.00%	91.14%	93.48%	92.00%
	95 cl [%]	88.75–99.17%	73.26–94.18%	83.78–95.34%	82.47–97.76%	85.78–96.10%
MMP-10	value [%]	78.67%	60.00%	74.68%	65.22%	71.20%
	95 cl [%]	67.68–87.29%	45.18–73.59%	67.31–80.86%	53.46–75.37%	62.42–78.95%
MMP-26	value [%]	85.33%	60.00%	76.19%	73.17%	75.20%
	95 cl [%]	75.27–92.44%	45.18–73.59%	69.23–81.99%	60.16–83.12%	66.68–82.49%
CA125	value [%]	97.33%	98.00%	98.65%	96.08%	97.60%
	95 cl [%]	90.70–99.68%	89.35–99.95%	91.29–99.80%	86.19–98.97%	93.15–99.50%
HE-4	value [%]	73.33%	94.00%	94.83%	70.15%	81.60%
	95 cl [%]	61.86–82.89%	83.45–98.75%	85.85–98.23%	61.60–77.49%	73.68–87.96%

**Table 6 ijms-26-05661-t006:** ROC curve analysis for the studied parameters in serous OC total group.

Parameters	AUC	SE_AUC_	95% CI	*p* (AUC = 0.5)
MMP-7	0.9566	0.01756	0.9222–0.9911	<0.0001
MMP-26	0.7550	0.04279	0.6711–0.8389	<0.0001
MMP-3	0.5821	0.04903	0.4860–0.6783	0.08502
MMP-10	0.7939	0.03749	0.7204–0.8673	<0.0001
CA125	0.9840	0.006660	0.9709–0.9971	<0.0001
HE4	0.9136	0.01971	0.8749–0.9522	<0.0001

Red indicates a statistically significant correlation.

**Table 7 ijms-26-05661-t007:** ROC curve according to FIGO classification by stage.

Stage I–II				
Parameters	AUC	SE_AUC_	95% CI	*p* (AUC = 0.5)
MMP-7	0.9548	0.01942	0.9167–0.9928	<0.0001
MMP-26	0.7555	0.04967	0.6581–0.8528	<0.0001
MMP-3	0.5004	0.05920	0.3844–0.6165	0.9943
MMP-10	0.7939	0.03749	0.7204–0.8673	<0.0001
CA125	0.9840	0.006660	0.9709–0.9971	<0.0001
HE4	0.9179	0.03020	0.8587–0.9772	<0.0001
**Stage III–IV**				
**Parameters**	**AUC**	**SE_AUC_**	**95%** **CI**	** *p* ** **(AUC = 0.5)**
MMP-7	0.9574	0.01791	0.9223–0.9926	<0.0001
MMP-26	0.7548	0.04513	0.6663–0.8433	<0.0001
MMP-3	0.5865	0.05051	0.4875–0.6855	0.08568
MMP-10	0.8065	0.03683	0.7343–0.8787	<0.0001
CA125	0.9955	0.003221	0.9892–1.002	<0.0001
HE4	0.6843	0.03817	0.6095–0.7591	<0.0001

Red indicates a statistically significant correlation.

**Table 8 ijms-26-05661-t008:** Characteristics of examined groups: High-Grade Serous Ovarian Carcinoma group of patients and control groups: benign ovarian tumour (cystis serous) patients and control group of healthy women.

	Studied Group	Number of Patients	Age (Median) (Range)	Group Size	
				Premenopausal	Postmenopausal
High-Grade Serous Ovarian Carcinoma in total	High-Grade Serous Ovarian Carcinoma in total	100	60 26–91	22	78
	Stage I	7	51 26–70	4	3
	Stage II	17	46 36–74	9	8
	Stage I–II	24	48 26–74	13	11
	Stage III	38	61 36–91	4	34
	Stage IV	38	64 34–80	5	33
	Stage III–IV	76	63 34–91	9	67
Control groups	Benign ovarian tumour (cystis serous)	73	48 25–80	37	36
	Healthy women	50	47 32–69	24	26

## Data Availability

Data is unavailable due to privacy or ethical restrictions.

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
