# Peer review of "Plasma Concentrations of Matrilysins (MMP-7, MMP-26) and Stromelysins (MMP-3, MMP-10) as Diagnostic Biomarkers in High-Grade Serous Ovarian Cancer Patients"

_ijms, 2025, doi:10.3390/ijms26125661_

Round 1
Reviewer 1 Report
Comments and Suggestions for Authors
The authors present a report on the diagnostic potential of MMPs in serous ovarian cancer. There are several scientific and technical concerns:
Major points:
- In this study, MMP-7 and MMP-26 emerge as the most powerful diagnostic biomarkers for ovarian cancer. However, these two biomarkers have already been investigated in the authors' previous studies. Therefore, the authors should clarify the novelty of the observation in this study.
- The authors compared the biomarkers among serous ovarian cancer (OC - total group), cysts (BL), and healthy women (HW) groups, but there is no HW group at all in some figures. Perhaps the authors have mislabeled HW as HS. In Table 8, all dots (.) are wrongly written as periods (,). Therefore, the authors should carefully check the data and their presentation.
- As shown in the ROC plots, the diagnostic performance of MMPs is comparable to or worse than the classical biomarker CA125. Therefore, the adding value of MMPs when used in combination with CA125 should be verified to prove the usefulness of MMPs.
- The authors did not test the influence of confounding factors such as infection, cardiovascular disease, age and menopausal status.
- Related to the previous point, the patient exclusion creteria are not clear.
Minor points:
- The color usage of ROC curves can be improved. The authors could use more discernable colors for different biomarkers.
- A few English phrases should be re-written for clarity. For example, the meaning of the phrases "unhelpful test" and "remains constant" is confusing.
Author Response
The authors present a report on the diagnostic potential of MMPs in serous ovarian cancer. There are several scientific and technical concerns:
Dear Reviewer.
We would like to thank You very much for Your thorough and honest review of our manuscript “Plasma concentrations of matrilysins (MMP-7, MMP-26) and stromelysins (MMP-3, MMP-10) as diagnostic biomarkers in serous ovarian cancer patients.” We will try to carefully answer all the Reviewer's questions and objections. Our answers will be written in green italics.
Major points:
- In this study, MMP-7 and MMP-26 emerge as the most powerful diagnostic biomarkers for ovarian cancer. However, these two biomarkers have already been investigated in the authors' previous studies. Therefore, the authors should clarify the novelty of the observation in this study.
Thank you for Your comment. In fact, in our previous studies we have determined MMP-7 and MMP-26, however these determinations were performed in a mixed group of ovarian cancer patients. In previous studies, patients with all histological subtypes of ovarian cancer - serous low and highly differentiated, endometroid, mucinous and clear cell carcinoma - were included in the study group. In the present study, we included only patients with low-differentiated serous carcinoma, which is the most common type of ovarian cancer, and also with the worst prognosis. Therefore, our group is very homogeneous and contains only one type of ovarian cancer, which affects the consistency of the conducted study, thus distinguishing it from previous studies of our team.
- The authors compared the biomarkers among serous ovarian cancer (OC - total group), cysts (BL), and healthy women (HW) groups, but there is no HW group at all in some figures. In Table 8, all dots (.) are wrongly written as periods (,). Therefore, the authors should carefully check the data and their presentation.
We sincerely appreciate the reviewer’s careful review and for pointing out these mistakes. We have corrected the figures accordingly, as well as the tables have been fixed.
- As shown in the ROC plots, the diagnostic performance of MMPs is comparable to or worse than the classical biomarker CA125. Therefore, the adding value of MMPs when used in combination with CA125 should be verified to prove the usefulness of MMPs.
Thank You for Your comment, this value will be added to the work as supplementary material.
- The authors did not test the influence of confounding factors such as infection, cardiovascular disease, age and menopausal status.
Thank You for Your comment. However, we would like to emphasize that all our patients were healthy and had no comorbidities. Because of that, we did not examine the effect of these disorders on marker concentrations. Age and menopausal status were included in the study however, since we did not show any significantly statistical correlations we will not present them in our paper.
- Related to the previous point, the patient exclusion creteria are not clear.
Thank You for Your comment. We have revised the description of the exclusion criteria to provide more detail.
Minor points:
- The color usage of ROC curves can be improved. The authors could use more discernable colors for different biomarkers.
Thank You for Your comment, the colors on the ROC curve have been corrected.
- A few English phrases should be re-written for clarity. For example, the meaning of the phrases "unhelpful test" and "remains constant" is confusing.
Thank You for Your comment. The phrases you pointed out, and more have been corrected.
Once again, we thank the reviewer for Their valuable guidance and corrections. We hope that the revised manuscript meets the reviewer’s expectations and will be suitable for publication in International Journal of Molecular Science.
Best regards,
prof. Sławomir Ławicki
also, on behalf of all authors
Reviewer 2 Report
Comments and Suggestions for Authors
Ovarian cancer is, as the authors state in the introduction, a silent killer, because it does not give any symptoms for a long time. Therefore, the search for markers, including markers that could be used for screening, is very much needed. Currently, diagnostics of many diseases are moving towards searching for molecular markers, e.g. specific genes, mutations, or microRNA levels. In the case of ovarian cancer, comprehensive genomic profiling is needed, assessment of homologous recombination status. Performing this test will significantly expand the group of patients with ovarian cancer who will be able to benefit from the introduction of appropriate treatment for them. Approximately half of patients who do not have BRCA1, 2 mutations have a deficiency of homologous recombination.
However, testing markers at the protein level, including enzymes from the MMP family, is also justified. MMPs are associated with neoplastic growth and a tendency to metastasize, but their main feature limiting their use as a marker is high non-specificity. Therefore, if they are to be used as markers, they should only be used in groups, in biochemical panels with other proteins.
Comments:
The work is relatively simple methodologically. It is limited to one biochemical method (ELISA) and the determination of selected MMPs. It is not fully known why these and not other MMPs were the subject of the analysis. Another limitation of the work is the lack of determination of MMP inhibitors - TIMPs, which inhibit the activity of MMPs in vivo. Determination of the concentration of inhibitors could be used to calculate MMP/TIMP ratios, which indirectly indicate the activity of specific MMPs in vivo.
The results are unnecessarily repeated 3 times: in the text, Table 1 and in the graphs. I propose to remove Table 1.
No indication of the static significance of results in Table 2.
I do not understand the need to perform the correlations placed in Table 3. For what purpose were the concentrations of individual MMPs correlated with each other? What new and clinically significant was behind such a correlation? In addition, despite several cases of obtaining statistically significant correlations, they are very small (they range from -0.3 to 0.3). Please explain these issue.
Table 4. No explanation in the legend for abbreviations placed in line 1? How were the values given in Tables 4 and 5 calculated?
Table 9. It is worth adding a simple comparison test, e.g. ch2, whether the difference between the number of women in the study group (ovarian cancer) and the control group (healthy and serous cyst together) before and after menopause was statistically significant. A cursory glance indicates that there were significantly more women with ovarian cancer in the study group than in the control group. Could this fact have had any influence on the results?
Author Response
Ovarian cancer is, as the authors state in the introduction, a silent killer, because it does not give any symptoms for a long time. Therefore, the search for markers, including markers that could be used for screening, is very much needed. Currently, diagnostics of many diseases are moving towards searching for molecular markers, e.g. specific genes, mutations, or microRNA levels. In the case of ovarian cancer, comprehensive genomic profiling is needed, assessment of homologous recombination status. Performing this test will significantly expand the group of patients with ovarian cancer who will be able to benefit from the introduction of appropriate treatment for them. Approximately half of patients who do not have BRCA1, 2 mutations have a deficiency of homologous recombination.
However, testing markers at the protein level, including enzymes from the MMP family, is also justified. MMPs are associated with neoplastic growth and a tendency to metastasize, but their main feature limiting their use as a marker is high non-specificity. Therefore, if they are to be used as markers, they should only be used in groups, in biochemical panels with other proteins.
Dear Reviewer.
We would like to thank You very much for Your thorough and honest review of our manuscript “Plasma concentrations of matrilysins (MMP-7, MMP-26) and stromelysins (MMP-3, MMP-10) as diagnostic biomarkers in serous ovarian cancer patients.” We will try to carefully answer all the Reviewer's questions and objections. Our answers will be written in green italics.
Comments:
The work is relatively simple methodologically. It is limited to one biochemical method (ELISA) and the determination of selected MMPs. It is not fully known why these and not other MMPs were the subject of the analysis. Another limitation of the work is the lack of determination of MMP inhibitors - TIMPs, which inhibit the activity of MMPs in vivo. Determination of the concentration of inhibitors could be used to calculate MMP/TIMP ratios, which indirectly indicate the activity of specific MMPs in vivo.
Thank You, for Your comment. First, we would like to emphasize that two methods were used in our study — ELISA and CMIA. The determination of the selected MMPs was conducted as a pilot study, aimed at identifying trends in the concentration changes of these molecules. Based on these preliminary findings, we plan to perform similar analyses in the future on a larger patient population and using additional research methods.
Second, the selection of the specific MMPs analyzed in this study was based on a thorough literature review and on our own previous measurements. The aim of the study was to evaluate the levels of matrilysins and stromelysins in serous ovarian cancer.
Third, as mentioned earlier, this is a pilot study, and we are therefore limited by the amount of data available. For this reason, TIMP levels were not assessed in the current work. However, we intend to include such measurements in future research and to calculate the MMP-to-TIMP ratio.
We hope for the reviewer’s understanding.
The results are unnecessarily repeated 3 times: in the text, Table 1 and in the graphs. I propose to remove Table 1.
Thank You, for Your comment. Table 1 has been removed.
No indication of the static significance of results in Table 2.
Thank You, for Your comment, statistical significance has been added to the table.
I do not understand the need to perform the correlations placed in Table 3. For what purpose were the concentrations of individual MMPs correlated with each other? What new and clinically significant was behind such a correlation? In addition, despite several cases of obtaining statistically significant correlations, they are very small (they range from -0.3 to 0.3). Please explain these issue.
Thank you very much for your valuable comment. The correlations presented in Table 3 were intended to explore potential relationships between the concentrations of individual MMPs in the context of ovarian cancer. Although the observed correlation coefficients are generally low, even such weak associations may reflect subtle co-regulatory mechanisms or shared biological pathways within the tumor microenvironment, such as those involved in extracellular matrix remodeling, angiogenesis, or metastasis.
We agree that these correlations alone do not have immediate clinical implications. However, they may offer preliminary insights into the complex biological interactions among MMPs in ovarian cancer and could serve as a foundation for future hypothesis-driven studies aimed at elucidating their functional roles.
Table 4. No explanation in the legend for abbreviations placed in line 1? How were the values given in Tables 4 and 5 calculated?
Thank You, for Your comment.. The abbreviations have been added below the table. The values presented in Tables 4 and 5 were obtained using MedCalc's free statistical calculators.
Table 9. It is worth adding a simple comparison test, e.g. ch2, whether the difference between the number of women in the study group (ovarian cancer) and the control group (healthy and serous cyst together) before and after menopause was statistically significant. A cursory glance indicates that there were significantly more women with ovarian cancer in the study group than in the control group. Could this fact have had any influence on the results?
Thank You, for Your comment. The Chi-square test will be included in the supplementary material.
We analyzed the effect of menopausal status on the obtained results; however, no statistically significant differences were observed. Therefore, considering the limited sample size, we decided not to include these results in the manuscript.
Once again, we thank the reviewer for their valuable guidance and corrections. We hope that the revised manuscript meets the reviewer’s expectations and will be suitable for publication in Cancers.
Best regards,
prof. Sławomir Ławicki
also, on behalf of all authors
Round 2
Reviewer 1 Report
Comments and Suggestions for Authors
The authors addressed most of my previous concerns, except for those regarding the ROC curves. The color usage of the revised ROC curves is still difficult to discern, with multiple curves with very similar color usage. There are sophisticated color palettes in public domain, such as the "Set2" or the "Paired" palettes in R, which have been described in public resources like https://www.datanovia.com/en/blog/top-r-color-palettes-to-know-for-great-data-visualization/. Additionally, the font used for the ROC curves should be made consistent as well.
Author Response
The authors addressed most of my previous concerns, except for those regarding the ROC curves. The color usage of the revised ROC curves is still difficult to discern, with multiple curves with very similar color usage. There are sophisticated color palettes in public domain, such as the "Set2" or the "Paired" palettes in R, which have been described in public resources like https://www.datanovia.com/en/blog/top-r-color-palettes-to-know-for-great-data-visualization/. Additionally, the font used for the ROC curves should be made consistent as well.
Thank you for your comment, the colors of the lines on the ROC curves have been corrected as suggested by the reviewer. We used the “Set2” color palette as suggested by the reviewer. The font for the ROC curves has also been improved.
Again, we thank the reviewer for all the guidance and corrections. It is our hope that the manuscript, after revision, will meet the reviewer's expectations and be published in the International Journal of Molecular Science
Best regards,
prof. dr hab. Sławomir Ławicki
also, on behalf of all authors
Reviewer 2 Report
Comments and Suggestions for Authors
The manuscript has been corrected by the authors, and the doubts present in the first version have been clarified.
Unnecessarily data have been removed, table legends have been clarified.
The MMPs selected for the study are consistent with the literature. The evaluation of less frequently studied MMPs (such as MMP-26 which has 117 articles in the PubMed database, for comparison MMP-3 has about 8500 titles) significantly expands the knowledge about the relationship between this group of enzymes in ovarian cancer.
I have no further comments.
Author Response
Thank You for the positive reception of our manuscript. Again, thank You for Your thorough and honest review and for Your time.
Best regards,
prof Sławomir Ławicki
also, on behalf of all authors